# Analysis of position and strength of westerlies and trades and their impact on the Agulhas leakage and the South Benguela upwelling

Nele Tim[1,2], Eduardo Zorita[1], Kay-Christian Emeis[2], Franziska U. Schwarzkopf[3], Arne Biastoch[3,4], and Birgit Hünicke[1]

[1]Helmholtz-Zentrum Geesthacht, Institute of Coastal Research, Geesthacht, Germany
[2]University of Hamburg, Institute for Geology, Hamburg, Germany
[3]GEOMAR Helmholtz Centre for Ocean Research Kiel, Kiel, Germany
[4]Kiel University, Christian-Albrechts-Platz 4, 24118 Kiel, Germany

**Correspondence:** Nele Tim (nele.tim@hzg.de)

**Abstract.** The westerlies and trade winds over the South Atlantic and Indian Ocean are important drivers of the regional oceanography around Southern Africa, including features such as the Agulhas Current, the Agulhas leakage and the Benguela upwelling. Agulhas leakage constitutes a fraction of warm and saline water transport from the Indian Ocean into the South Atlantic. The leakage is stronger during intensified westerlies. Here, we analyze the wind stress of different observational and modelled atmospheric data sets (covering the last two millennia, the recent decades and the 21st century) with regard to the intensity and position of the south-easterly trades and the westerlies. The analysis reveals that variations of both wind systems go hand in hand and that a poleward shift of westerlies and trades and an intensification of westerlies took place during the recent decades. Furthermore, upwelling in South Benguela is slightly intensified when trades are shifted poleward. Projections for strength and position of the westerlies in the 21st century depend on assumed $CO_2$ emissions, and on their effect relative to the ozone forcing. In the strongest emission scenario (rcp8.5) the simulations show a further southward displacement, whereas in the weakest emission scenario (rcp2.6) a northward shift is modelled, possibly due to the effect of ozone recovery dominating the effect of anthorpogenic greenhouse forcing. We conclude that the Agulhas leakage has intensified during the last decades and is projected to increase if greenhouse gas emission are not reduced. This will have a small impact on Benguela upwelling strength, and may also have consequences for water mass characteristics in the upwelling region. An increased contribution of Agulhas water to the upwelling water masses will import more preformed nutrients and oxygen into the upwelling region.

## 1 Introduction

The regional oceanographic phenomena around Southern Africa, the Agulhas Current, the Agulhas leakage, and the Benguela upwelling, are all three to a large extent influenced by one of the two wind regimes in this region, the Southern Hemisphere westerlies and the easterly trade winds. In this study we analyze several observational data sets and model simulations to

understand the variability and trends in the intensity and spatial distribution of westerly and trade winds in the South Atlantic-Indian Ocean over the past few decades, the last century and past two millennia. We also compare observed trends with trends projected in future climate simulations driven, among others, by changing greenhouse gas and ozone concentrations.

The Benguela upwelling system is located off the southwest coast of Africa (Blanke et al., 2005). This coastal upwelling system is one of the four eastern boundary upwelling systems of the world (California, Humboldt, Canary, and Benguela) (Shannon, 1985). These upwelling systems, which are major areas of ocean primary production, are driven by the trade wind system of the subtropics and tropics. The equatorward wind forces surface water along the coast to move offshore. Cold and nutrient-rich water wells up into the sunlit layer near the coast due to mass balance (Bakun et al., 2010). Therefore, these regions have

characteristic sea surface temperature (SST) patterns and are exceptionally productive ecosystems.

The Benguela upwelling system is divided into two distinct sub-systems, North and South, separated by the Lüderitz upwelling cell at 27° S (Hutchings et al., 2009). Upwelling in South Benguela varies seasonally with strongest upwelling in austral summer (December-February) (Tim et al., 2015), when the South Atlantic subtropical high has moved southward to induce trade winds parallel to the coast. Upwelling feed water in the South Benguela upwelling region is the Eastern South Atlantic Central

Water (ESACW) that mixes in the Cape Basin with Indian Ocean water imported by the Agulhas leakage (Mohrholz et al., 2008; Garzoli and Gordon, 1996). The strength of Agulhas leakage dynamically influences the South Benguela upwelling: larger portions of warm saline Indian Ocean water may cause enhanced stratification and weaken the upwelling. In addition, in terms of nutrient and $CO_2$ budgets, water entering through the Agulhas leakage significantly contributes preformed nutrients that contemporaneously render South Benguela into a sink for atmospheric $CO_2$ (Emeis et al., 2018). Concerning centennial

variability of upwelling over the period 1600-1900, Granger et al. (2018) reasoned from their analysis of grain size and SST changes derived from marine sediment records that the inflow of Agulhas water into the South Benguela upwelling system is stronger when westerlies are located further to the south and SSTs are cooler during northernmost positions of the westerlies.

 Just south of the upwelling region, Indian Ocean waters enter the South Atlantic as Agulhas Leakage, mainly transported by so called Agulhas rings. These eddies are shed when the Agulhas Current, the western boundary current of the South Indian

Ocean (Beal and Bryden, 1997) retroflects into the Indian Ocean at the southern tip of Africa. The resulting inter-oceanic connection is an important feature of the global meridional overturning circulation (Gordon, 1986).

The Agulhas leakage has undergone changes in conjunction with changes in the global climate. During glacial periods the leakage was strongly diminished, based on qualitative reconstructions of foraminiferal assemblage counts, whereas the transport of Indian Ocean waters into the South Atlantic was enhanced during interglacial periods (Peeters et al., 2004). On shorter and

more recent time scales, the leakage has been found to increase in ocean simulations over the second half of the 20th century driven by observed atmospheric forcing (Schwarzkopf et al., 2019; Biastoch et al., 2009). This intensification of the leakage can be attributed to an intensification of the Southern Hemisphere winds (Durgadoo et al., 2013). The impact of the position of the westerlies on the Agulhas leakage is still under debate. The studies of Ruijter (1982) and of Biastoch et al. (2009) found that a more northerly position hinders the westward flow of Agulhas water into the South Atlantic, whereas a more southerly

position leads to a wider passage of throughflow between the southern coast of South Africa and the westerlies. However,

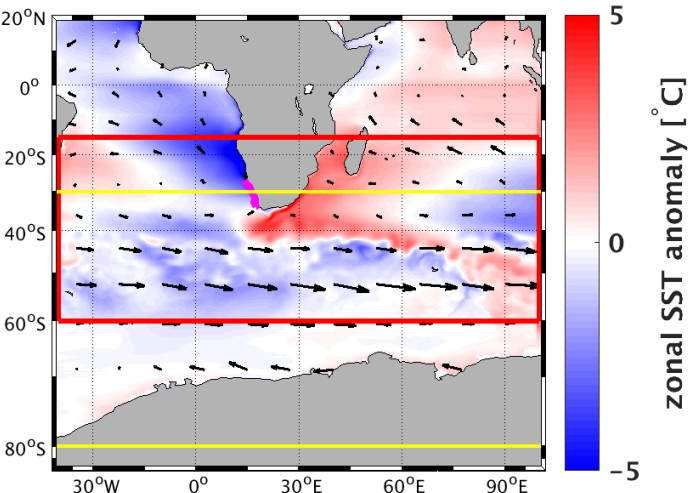

**Figure 1.** Figure of the research area. Shaded colors (deviations of sea surface temperature from their zonal mean of the INALT20 simulation) depict the warm Agulhas Current at the southeast coast of South Africa and the cool Benguela current at the southwest coast. Arrows show the mean wind stress field in the study region, derived from the COREv2 data set. The area between 40° W - 100° E, 15° S - 60° S in which the position and intensity of the trade winds and westerlies are calculated is indicated by the red box. The yellow lines represent the northern and southern border (30° S and 80° S) of where the Southern Hemisphere temperature gradient is calculated. The magenta line is the area where the upwelling index for South Benguela is taken (27° S-34° S, 100 km offshore to coast).

recent studies found diverging results. De Boer et al. (2013) showed that there is no linkage between the position of the zero wind stress curl and of the subtropical front. The study by Durgadoo et al. (2013) found that an enhanced leakage tends to occur when westerlies are shifted equatorwards due to the redistribution of momentum input by the winds.

Nevertheless, there is consensus on the impact of the westerlies on the Agulhas leakage, with the strength of westerlies being

the key driver of the leakage: stronger westerlies lead to a stronger wind stress curl and an intensified transport from the Indian Ocean    into    the    South    Atlantic    (Cheng    et    al.,    2018;    Durgadoo    et    al.,    2013).

Given the importance of variations in the position and strength of the wind systems in this region (westerlies and trade winds as well as wind stress curl), one goal of our study is to understand the consequences of changes in the wind system for the South

Benguela upwelling system. The direct impact of changes in the trades has been analyzed in a previous study (Tim et al., 2016, 2015). Here, we are interested in the indirect impacts on the South Benguela upwelling via changes in the westerlies which in turn impact the Agulhas leakage and thus the water masses in the Benguela upwelling region. Therefore, we search for a connection between position and strength of the wind systems that may explain synchronous variability and common trends of the Agulhas Current, leakage and Benguela upwelling. We then examine future trends in the wind system as simulated in future

climate simulations to reveal common trends in these three oceanic current systems. The analysis encompasses correlations of

position and strength of westerlies and trades, their trends, and the connection to the South Benguela upwelling. One drawback of the presently available climate simulations for this analysis, in particular for the past 2000 years and the 21st century, is their relatively coarse spatial resolution. A realistic representation of the impact of the wind stress on the oceanic circulation systems in this region may require models with finer spatial resolution. Acknowledging this unavoidable limitation, given the

present generation of global climate models, this type of analysis may still be valuable to understand the variability of the wind systems in this region and its possible drivers, guiding future studies based on more realistic models. Our present study can be considered as part of the ongoing comprehensive analysis using state-of-the-art climate models (e.g. Small et al., 2015; Wang et al., 2014).

## 2   Data and methods

For the analysis of the wind stress we use several different atmospheric data sets, all gridded and derived from simulations with atmospheric models, some of them with data assimilation. To investigate variations in the last six decades we use the NCEP/NCAR reanalysis 1 (National Centers for Environmental Prediction/National Center for Atmospheric Research Reanalysis version 1, hereafter NCEP1) (Kalnay et al., 1996), COREv2 (the Version 2 Forcing for Coordinated Ocean-ice Reference Experiments) (Large and Yeager, 2009), JRA-55 (the Japanese 55-year Reanalysis) (Kobayashi et al., 2015), and ECHAM6XR

(the atmospheric general circulation model of the Max Planck Institute for Meteorology with extra-high-resolution) (Schubert-Frisius et al., 2016). NCEP1 is analyzed here over the period 01/1948-08/2012 and has a horizontal resolution of T62 (200 km). The data set COREv2 covers the period 01/1958-12/2009 with the same horizontal resolution. The atmospheric state is given by NCEP1, whereas radiative fields, precipitation, sea-ice and SST are derived from other sources. The Japanese reanalysis data set JRA-55, analyzed for the period 01/1958-12/2013, has a resolution of T319 (about 55 km). The global data set

ECHAM6XR covers the period 01/1948-04/2015. It is the result of an atmosphere-only simulation with the spectral model ECHAM6. This model was nudged towards the NCEP1 reanalysis at spatial scales larger than about 1000 km (spectral nudging, von Storch et al. (2000)), whereas shorter spatial scales are allowed to evolve freely. Its spatial resolution is much finer than the NCEP1 reanalysis (T255, about 50 km).

In addition to the reanalysis data sets that cover approximately the last 60 years ($\sim$1950-2010), we include the ERA20C re-

analysis data set (European Centre for Medium-Range Weather Forecasts re-analysis for the 20th century) (Poli et al., 2016) to cover a longer time period from the last century (01/1900-12/2010) with a spatial resolution of T159 (125 km).

We additionally analyze two simulations with the Max-Planck-Institute Earth System Model (MPI-ESM-MR) (Giorgetta et al., 2013) for the historical period (1850-2005). The two simulations only differ in their initial conditions. We analyze three future scenarios (MPI-ESM-LR) with different strength in greenhouse gas forcing, representative concentration pathways rcp2.6,

rcp4.5, and rcp8.5, where the numbers indicate the anthropogenic radiative forcing in $Wm^{-2}$ reached by the year 2100 (Taylor et al., 2012). Tropospheric ozone concentrations are the same for all three rcp scenarios (Giorgetta et al., 2013). Furthermore, we analyze the variations of the wind stress during the last two millennia in a simulation with the Earth System Model MPI-ESM-P (ECHAM2k) (S. Wagner, pers. comm.).

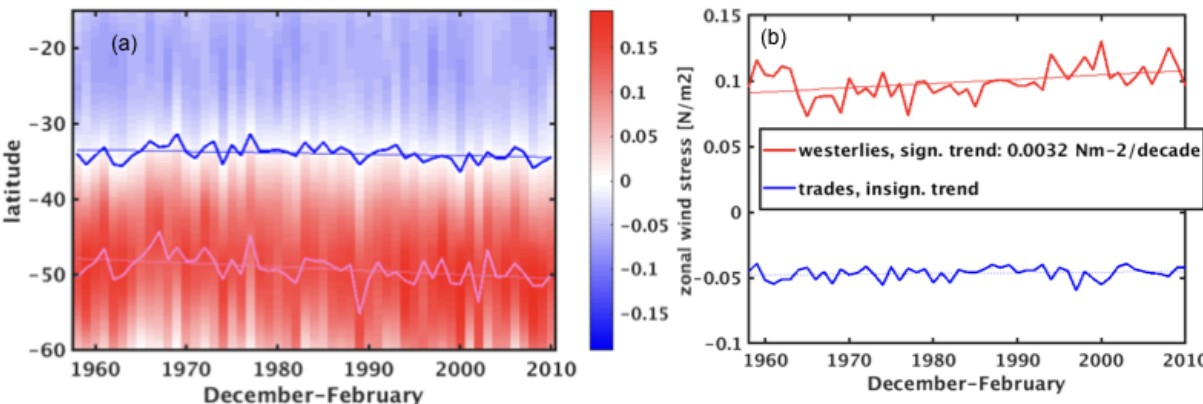

**Figure 2.** Temporal evolution of zonal mean (40° W-100° E) zonal wind stress [N/m2] (shaded color) and the meridional position of maximum of westerlies (light pink curve) and of wind stress curl (blue curve). The linear trends in their positions are also displayed. There is a significant trend of -0.52 °/decade in the position of the westerlies and an insignificant trend for the trades (a) and the temporal evolution of the intensity of wind stress of westerlies (mean between 60° S and of the latitude where wind stress turns eastward) and trades (mean between latitude where wind stress turns eastward and 15° S) and their corresponding linear trends (b), both in December–February derived exemplary from the COREv2 data set in the period 1958-2009.

For our study of the trade winds and the westerly wind band we analyze the averages of the austral summer season (December-January-February; DJF) of zonal wind stress over the whole South Atlantic and South Indian Ocean (40° W - 100° E, 15° S - 60° S, Fig. 1 red box). From these data, we calculated the temporal variation of the latitude of maximum westerly winds (hereafter, position of westerlies) and the latitude of maximum wind stress curl (the latitude where winds change from wester-
lies to easterly trades, hereafter position of trades). Furthermore, we calculated the intensity of the westerlies as the meridional mean between 60° S and the latitude where winds change from westerlies to trades to analyze the intensity of the wind stress in addition to the position of the wind systems. For the trades, we calculated the meridional mean between the latitude where winds change from westerlies to trades and 15° S. We focus on DJF because it is the season of maximum upwelling in South Benguela (Tim et al., 2015). To investigate whether there is an impact of El Niño-Southern Oscillation (ENSO), we used the
Multivariate-ENSO-Index (MEI, Wolter and Timlin (1993)), a bimonthly time series (1950–2009) including not only the SSTs and the sea level pressure (SLP), but also additional atmospheric variables for calculating the index. For ENSO, the seasonal mean of DJF was calculated by averaging the bimonthly means of December-January and January-February.

Vertical velocity in the ocean as simulated by the global ocean simulation INALT20 (1/20°horizontal resolution) driven by COREv2 atmospheric forcing (Schwarzkopf et al., 2019) is used as an upwelling index. The data was selected at 100 m depth,
in the region between 27° S-34° S, over a corridor of 100 km along the coast from the Cap Basin (34° S) to Lüderitz (27° S, see Fig. 1 magenta area).

For the statistical significance of the linear trends in the wind stress position and intensity and the linear correlations a significance level of p = 0.05 was adopted.

For the COREv2 data set we used the wind stress as seen by INALT20: this wind stress data set is the COREv2 data interpolated to the global host grid of INALT20, at 1/4° resolution and considering ocean velocities (relative winds). Since the position and intensity of the data sets COREv2, NCEP, and ECHAM6XR are significantly correlated (between r=0.85 and r=0.94), taking the ocean velocity into account when calculating the wind stress does not impact the conclusions of this study.

## 3   Results

### 3.1   Position and intensity of westerlies and trades

The position of westerlies and trades (DJF means) over the South Atlantic and Indian Ocean are calculated based on the wind stress of the data sets NCEP1, COREv2, JRA-55, ERA20C, and ECHAM6XR. The interannual variation in the positions of these two variables are significantly correlated with correlation coefficients between r=0.65 and r=0.78. Figure 2 (a) exemplarily shows the results of the COREv2 data set. For the other data sets, see Fig. S1.

The width of the westerlies band is defined here as the distance between the latitude of the maximum of the westerlies and the equatorward latitude where the wind stress changes to easterly trades. This width is negatively correlated to the position of the westerlies. Thus, a more southerly latitude of the maximum winds tends to occur simultaneously with a broader subtropical belt of westerlies. Both trades and westerlies tend to shift poleward or equatorward simultaneously, but the amplitude of westerlies displacement is stronger, leading to a correlation between the latitudinal position of the wind systems and the width of the subtropical belt of westerlies. All five data sets show highly significant correlations (r=-0.60 - r=-0.84). Figure 3 shows the wind stress of the COREv2 data set for the year 1964/1965 at the beginning of the time series (a) and in the year 2007/2008 at the end of the analysis period (b). The intensification, poleward displacement and widening are clearly borne out by these results.

We also investigate the variations in the intensity of the westerlies and the trade winds. Intensity of the westerlies is defined here as the zonal wind meridionally averaged between 60° S and the latitude of maximum wind stress curl; intensity of the trade winds is defined as the zonal wind meridionally averaged between the maximum wind stress curl and 15° S. As expected, the intensity and position of the westerlies are highly negatively correlated (r=-0.56 - r=-0.78), so that a more southerly position of the wind maximum is accompanied by stronger westerlies, and vice versa. Furthermore, there is also a positive correlation between the width of the westerlies band and the intensity of the westerlies, so that the width of the westerly band (equatorward of the maximum) is broader when westerlies are stronger (for ERA20C r=0.32 and ECHAM6XR r=0.45).

The correlation of the intensity of westerlies and trades is, although statistically not significant for NCEP1 and COREv2, negative (between r=-0.21 and r=-0.62). These negative correlations indicate a tendency of both wind systems to intensify or weaken at the same time, especially in the model simulation ECHAM6XR and the reanalyses data sets with higher spatial resolution than NCEP1 and COREv2. The correlations are numerically negative because winds are positive when directed eastward. Figure 2 (b) exemplarily shows the results of the COREv2 data set. For the other data sets, see Fig. S2.

Thus, both subtropical wind systems tend to vary coherently and a more poleward position goes with stronger westerlies and a

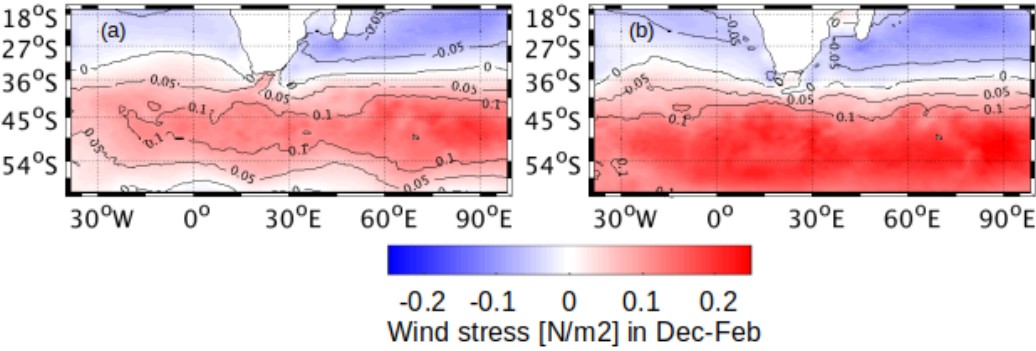

**Figure 3.** Temporal mean of the December–February wind stress [N/m2] of the periods 1964/1965 (a) and 2007/2008 (b) derived from the COREv2 data set. Contours and shading represent the same variable.

broader subtropical belt of westerlies, a more equatorward position goes with weaker and narrower belt of westerlies.

A trend analysis over the recent decades (1948-2012, 1958-2013, 1958-2009, and 1948-2015) reveals a shift of both westerlies and trades to more poleward positions in all data sets (Table 1). Trends in the position of the westerlies are stronger than those of the trades, and are statistically significant in all but one data set (position of trades in COREv2). The strength of the westerlies is also increasing; all data sets provide significant positive trends. The intensity of the trade winds shows a significant (positive) trend in JRA-55 and ECHAM6XR.

The reanalysis covering the whole last century, ERA20C, shows significant trends for position (poleward shift) and intensity (strengthening) of both wind systems, westerlies and trades.

The poleward shift and intensification of the westerlies is linked to trends in the Southern Annular Mode (SAM). This index is defined as the difference of SLP anomaly between 40°S and 60°S, here calculated from the COREv2 data set. Correlations reveal a strong and significant link between SAM and the position of the trades (r=-0.79) and westerlies (r=-0.73), as well as with the intensity of the westerlies (r=0.85). The trends identified in the wind systems as previously described are related to the positive trend in SAM over the recent decades (1958-2009). These results agree with the study of Loveday et al. (2015). They found that SAM modulates Indian Ocean westerlies and further detected the impact of this connection to the Agulhas leakage.

Expanding our research area to the whole Southern Hemisphere, again using the wind stress over the ocean between 15° S - 60° S derived from the NCEP1 and COREv2 data sets, reveals the comparable correlations and trends in both data sets as in the

**Table 1.** Trends of the position ($^\circ$/decade) and intensity (Nm-2/decade) of westerlies and trades for the five observational based data sets (NCEP1, COREv2, JRA-55, ERA20C, ECHAM6XR), for two free-running MPI-ESM simulations (r2 and r3) for the historical period, and for two simulations (r2 and r3) for three future scenarios. Statistically significant trends at the 95 % level are marked with an asterisk.

| Data set | position of westerlies | position of trades | intensity of westerlies | intensity of trades |
|---|---|---|---|---|
| Observational based/driven data sets: | | | | |
| NCEP1 | −0.61* | −0.23* | +0.005* | +0.0002 |
| COREv2 | −0.52* | −0.19 | +0.0032* | +0.0005 |
| JRA-55 | −0.50* | −0.27* | +0.005* | −0.001* |
| ERA20C | −0.34* | −0.19* | +0.004* | −0.001* |
| ECHAM6XR | −0.66* | −0.31* | +0.006* | −0.002* |
| MPI-ESM-MR historical period 1850-2005: | | | | |
| historical r2 | −0.005 | −0.004 | +0.0003 | +0.0002 |
| historical r3 | +0.008 | +0.003 | +0.0005* | none |
| MPI-ESM-LR future scenarios 2006-2100: | | | | |
| rcp2.6 r2 | +0.04 | +0.05 | −0.0009* | +0.0004* |
| rcp2.6 r3 | +0.14* | +0.07 | −0.0005 | +0.0003 |
| rcp4.5 r2 | +0.02 | −0.009 | +0.0001 | +0.0003 |
| rcp4.5 r3 | +0.05 | +0.02 | +0.0002 | +0.0004 |
| rcp8.5 r2 | −0.17* | −0.13* | +0.001* | +0.0003 |
| rcp8.5 r3 | −0.18* | −0.13* | +0.002* | +0.0004* |

previous analysis that was limited to the Atlantic and Indian Ocean sector. The position and intensity of the westerlies display significant trends, whereas the trades do not.

A candidate forcing mechanism for the trends and variability of the westerlies may involve the latitudinal surface temperature gradient (via the thermal wind equation). To test this hypothesis, we calculate the Southern Hemisphere meridional temperature gradient between 30 ° S and 80° S from the NCEP1 data (Fig. 1 yellow lines). This gradient was estimated with a linear fit of the zonal mean temperature against latitude for each December-to-February season, thus yielding one value of the gradient per year. Correlations between the latitudinal temperature gradient at the surface and the wind stress reveal a strong temperature gradient being associated with a southerly position of the westerlies (r=-0.83), a broader westerlies band (r=0.65) and stronger westerlies (r=0.8). The estimated impact of the meridional temperature gradient on the zonal wind stress profile between 45° S and 65° S is depicted in Fig. 4, along with the climatological meridional profile of the wind stress. The impact of the temperature gradient is estimated from a linear regression between the meridional temperature gradient in this region (predictor) and the zonal wind stress at each latitude band (predictand):

$$wind\_stress(latitude, time) = wind\_stress\_climatology(latitude) + \alpha * temperature\_gradient(time) + residual(latitude, time)$$

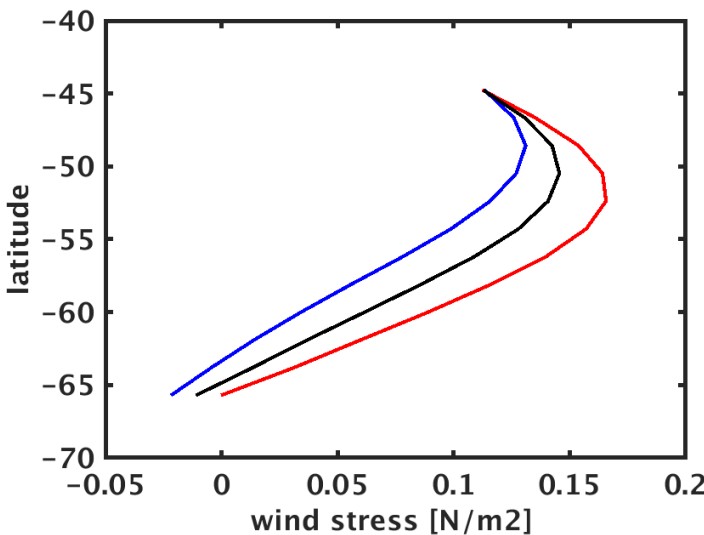

**Figure 4.** Impact of the meridional surface temperature gradient between 30° S - 80° S on the Southern Hemisphere wind stress intensity and position. Black curve shows the climatological mean of the zonal wind stress at the latitudinal range 45° S - 65° S in December–February (seasonal mean over the period 01/1948-08/2012) with NCEP1. The blue and red curves show the mean wind stress profile in years with a steeper temperature gradient (more than one standard deviation) (red line) or in years with a flatter temperature gradient (more than one standard deviation (blue line). The meridional temperature gradient was estimated as the slope of a linear fit of the zonal mean temperature against latitude.

(1)

Figure 4 shows with red and blue lines the mean wind stress profile in years with anomalous meridional temperature gradient: red, steeper than one standard deviation, blue flatter than one standard deviation.

## 3.2 Variability in the past and in future climate

The global ECHAM2k simulation is a coupled atmosphere-ocean simulation with a state-of-the art Earth System Model driven by external climate drivers (solar variations, volcanic activity and greenhouse gases). The simulation covers the last two millennia, for which statistical analyses of trends and variability of the wind systems were conducted similarly to those for reanalysis data in the previous section. In general, the correlations of position and intensity of trades and westerlies support the inferences based on the reanalysis data sets and the global downscaling product ECHAM6XR. Variations in the latitudinal position and strength of westerlies and trades tend to be simultaneous and the westerlies band is broader when the westerly maximum has a more poleward position. Regarding centennial trends, the results are more ambiguous: calculated for each century separately, they suggest that winds are stronger when located more poleward in 16 (14) centuries out of 21 for the westerlies (trades) (Fig. S3). During the 16th century, a particularly cold century during the Little Ice Age (LIA) between the 15th and the 19th

centuries, westerlies significantly weakened and shifted equatorward. This is supported by the studies by Hahn et al. (2017), Stager et al. (2012), and Granger et al. (2018), who also found that the winds shifted equatorward during cooler and poleward during warmer periods. Nevertheless, the position of Southern Hemisphere westerlies during the LIA is still under debate and seems to depend on the region, as studies of the African, the South American and the Australian sector provide varying results (Chase and Meadows, 2007).

The MPI-ESM simulations for the historical period (1850-2005) and the future (2006-2100) reveal the same coherence of westerlies and trades as the other data sets (NCEP1, COREv2, JRA-55, ERA20C, and ECHAM6XR). However, a significant difference is that the intensity of trades and westerlies are not correlated in the the rcp4.5 and rcp8.5 simulations. Also, the trends for most parts are statistically not significant over the historical period and only partly significant in scenarios of future radiative forcing (Table 1). Although the simulations of the historical period produce weaker trends than observed, we here provide an explanation as to why the simulations with different scenarios of greenhouse emissions produce different trends in the wind systems. This explanation involves the compensation of diverging trends caused by ozone on the one hand and by anthropogenic greenhouse gases on the other hand. The strongest scenario rcp8.5 indicates a poleward shift and intensification of westerlies (and a poleward shift and weakening of the trades). In the weaker emission scenario (rcp2.6), by contrast, the simulation displays a northward shift of the westerlies and a weakening of trades and westerlies. As prescribed ozone concentrations are the same in all three scenarios, the amount of anthropogenic greenhouse gas emissions is likely the factor that causes the difference in the simulated trends in the wind systems. It has previously been found in simulations that the ozone recovery causes a northward shift and a weakening of the tropospheric jet and a lowering of the SAM (Southern Annular Mode) values (Watson et al., 2012). This is likely the effect seen in the simulations with the weaker rcp2.6 scenario. Only with the stronger rcp8.5 scenario the emissions are strong enough to counteract the effects of ozone recovery. The simulated trends under rcp4.5 forcing are not statistically significant, which in our interpretation would indicate a balance of both driving factors, ozone recovery and anthropogenic greenhouse gas emissions. It has to be kept in mind that details of these results depend on the model.

The comparison of the time series of the COREv2 to the MPI-ESM historical runs in the period 1958-2009 reveal that the position of the westerlies is more to the north during the last 20 years in the coupled model simulation than in the reanalysis data set (Fig. 5) and at a similar latitudinal position as the reanalysis in the beginning of the simulation time period. This leads to the not significant trends in the historical period in the MPI-ESM simulations. Therefore, the response of the wind stress to the increase in anthropogenic greenhouse gases and decrease in ozone is weaker in the MPI-ESM simulation than in the observational based COREv2 data. This weaker sensitivity, detected in the historical period, may be relevant for the real future trends. Thus, the simulated decrease of the westerlies in the scenario rcp2.6 and its increase under scenario rcp8.5 may be an underestimation of the expected response.

### 3.3 The impact of the position of westerlies and trades on the South Benguela upwelling

For this section we limit our analysis to the COREv2 data set and use the simulated vertical velocity of the COREv2-driven INALT20 ocean simulation to investigate the link between the westerlies and upwelling intensity in the Benguela upwelling

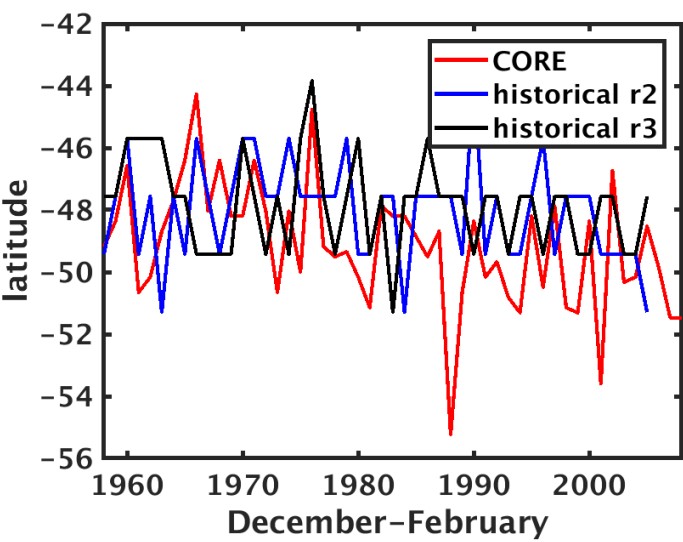

**Figure 5.** Meridional position of the maximum intensity of westerlies in December–February derived from the COREv2 reanalysis and the MPI-ESM historical simulations r2 and r3.

system. The correlation between the upwelling (simulated vertical velocity at 100 m depth), spatially averaged over the South Benguela upwelling region (27° S-34° S, 100 km width band off the coast, Fig. 1 magenta area), and the position of the trades is not significant, but shows a weak tendency towards stronger upwelling when the trades are located further south. In contrast to their position, the intensity of the trades does show a significant negative correlation with upwelling (r=-0.33), implying
stronger upwelling when trades are intensified.

Although the link between the position of the trades and upwelling is weak, there is a clear correlation between position of the trades and SSTs (Fig. 6). A northerly position of the trades is linked to negative SST anomalies west of the southern tip of the African continent and in the offshore part of the South Benguela upwelling region, but not directly at the coast. The correlation of the position of the trades and the spatial average of the SST over the upwelling region is significant and positive (r=0.43).
Thus, a northerly position of the trade winds is associated with warm SSTs in the upwelling region and with cooler SSTs further offshore (Fig. 6). This is explained by the relationships between the position of the trades and intensity of upwelling on the one hand, and by the relationship between the position of the trades and the Agulhas leakage on the other hand. First, warm SSTs in the upwelling region indicate that upwelling is weakened during a more northerly position of the trades. Secondly, the Agulhas leakage is weakened during weaker westerlies. Since the position of the trades and the westerlies is significantly correlated, a
more northerly position of the trades leads to weaker westerlies. Furthermore, since westerlies and trades in general shift into the same direction, the low SST offshore of the upwelling on the shelf are caused by a weak Agulhas leakage and a reduced transport of warm water from the Indian Ocean into the Cape Basin.

 ENSO has been found to have some influence on the South Atlantic trade winds, on the westerlies and also on the Benguela upwelling (Tim et al., 2015; Philippon et al., 2012). The correlation between ENSO and the trades is relatively strong and

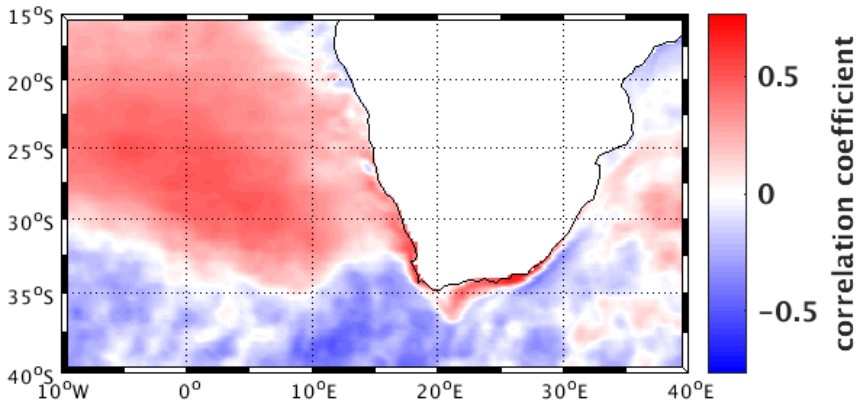

**Figure 6.** Correlation pattern of the sea surface temperature in the INALT20 simulation and the position of the trades of COREv2 reanalysis in December–February over the period 1958-2009.

significant (intensity: r=0.38, position: r=0.46), whereas the correlation between ENSO and the westerlies is weak and not significant. Nevertheless, the sign of both correlation is the same, so that ENSO does not disrupt the tendency of the trades and westerlies to intensify or weaken at the same time. The position of trades is displaced to the north, trades are weak during a positive ENSO phase (El Niño) and Benguela upwelling is slightly reduced (Tim et al., 2015). Westerlies also tend to be

located further north during an El Niño event (although this link is statistically weaker) and the Agulhas leakage is reduced, leading to cooler SSTs in the Cape Basin and offshore of the upwelling region.

## 4 Discussion and conclusions

We analyzed the intensity and position of the trade winds and of the belt of westerlies in the South Atlantic and South Indian Ocean with regard to implications for the intensity of the Agulhas leakage and the South Benguela upwelling, during austral

summer (December-February) as modelled for the last two millennia, the last century, the past 60 years ($\sim$1950-2010) and projected for the 21st century.

Our conclusions are listed and discussed in the following:

- *Link between position and intensity of trades and westerlies:* The analysis of the reanalysis data sets NCEP1, COREv2,

JRA-55, and ERA20C, the global high-resolution simulation ECHAM6XR and the Earth System Model MPI-ESM

shows that interannual latitudinal shifts in the position and intensity of trades and westerlies go hand in hand. Both systems tend to simultaneously shift latitudinally. When the shift is poleward they also tend to become stronger. These results are confirmed by the analyses of the position and intensity of the wind stress over the entire southern ocean between 60°S and 15°S. One driving mechanisms appears to be the surface air temperature difference between the subtropics and midlatitudes, as position and intensity of the wind systems are correlated with the meridional temperature gradient: the stronger the gradient, the further south and stronger are the westerlies. As for these atmospheric data sets that are generated in simulations where the SSTs are prescribed, the causal link between the temperature gradient and the wind stress can only come from the prescribed SSTs to the simulated wind stress. Thus, these correlations indeed result from a physical link. In the case of coupled simulations, the answer is not as clear. However, there is a very plausible mechanism by which temperature gradients drive the geostrophic part of the westerlies through the effect of temperature on density. A mechanism by which zonal winds may physically cause meridional temperature gradients is not as plausible.

Correlations with the Southern Annular Mode (SAM) reveal a strong connection to the sea level pressure (SLP) difference between the subtropics and midlatitudes as well.

– *Trends over the historical period*: During the recent decades the westerlies have shifted poleward and have become more intense (in the reanalysis data sets). However, the trends of the MPI-ESM for the historical time period (1850-2005) are weaker than in the reanalysis data sets and mainly insignificant (even when analyzing a time span comparable to that covered by the other data sets).

Swart and Fyfe (2012) also found an intensification and poleward shift of the Southern Hemisphere surface westerly wind stress jet in various Coupled Model Intercomparison Project model simulations (CMIP3 and CMIP5) and reanalysis data sets. Furthermore, there are reasons that suggest that the trends in the reanalysis are not artifacts. NCEP1 has been shown to have an above-average trend of SAM (Marshall, 2003) and, therefore, probably in other parameters dependent on SLP in that region. COREv2 is based on NCEP1. The global simulation with the atmospheric general circulation model ECHAM6XR is only dependent on NCEP1 and shows even stronger trends than the reanalysis data sets. Also analyzing the more recent data sets JRA-55 and ERA20C confirms our results of a poleward shift and intensification of the westerlies and the trades over the South Atlantic and Indian Ocean. Thus, these long-term trends are present across different reanalysis data sets that assimilate different observations.

Watson et al. (2012) and McLandress et al. (2011) found that increasing anthropogenic greenhouse gas emissions and the ozone depletion drive the trends in position and intensity of the westerlies over the past decades. The ozone concentration has not fully recovered yet from the minimum attained in the 1990s, and a slightly positive trend in observations since the year 2000 is overlain by large interannual variability (Solomon et al., 2016). The trends in the intensity and position of the westerlies have been found to cause an intensification of the Agulhas leakage in ocean simulations driven by atmospheric meteorological reanalysis (Biastoch et al., 2009). Durgadoo et al. (2013) and Loveday et al. (2014) showed that Agulhas leakage and Agulhas Current are decoupled, but that both respond to the intensification of westerlies

and trades, respectively. Beal and Elipot (2016) confirm that intensified winds impact the Agulhas Current, though not by strengthening it but by broadening it. Thus, the coherent variability and trends in the two wind systems cause a modulation of both oceanic components, Agulhas Current and Agulhas leakage.

– *Link between wind systems, Agulhas leakage, and South Benguela upwelling:* The meridional shifts of the trades and the westerlies and the associated variations in their intensity has an impact on the SSTs around Southern Africa. The position of the winds is only weakly correlated with South Benguela upwelling intensity (stronger upwelling when trades are shifted poleward). In contrast, the strength of the trades is significantly correlated with Benguela upwelling, with more intense trades being linked to stronger upwelling in South Benguela. Furthermore, a more southerly position of westerlies and trades leads to positive SST anomalies in the Cape Basin and the offshore part of the South Benguela upwelling region. This SST anomaly pattern cannot be explained by a more intense upwelling, which is usually linked to more intense trades and westerlies. Instead, the physical mechanisms linking variations in strength and position of the wind systems to SST in the South Atlantic is the Agulhas leakage, which modulates the advection of warm Indian Ocean water into the South Atlantic and into the Benguela upwelling system.

This agrees with proxy-based findings of Granger et al. (2018) for the late-Holocene. They found that during a northward shift of the westerlies, SSTs have been lower in South Benguela, not caused by intensified upwelling but due to changes in the advected water masses.

– *Trends over the last two millenia:* Regarding the centennial time scales over the past two millennia, the analysis of the wind stress variations (of the general circulation model ECHAM2k) is consistent with the results obtained for the observational period. Both wind systems, westerlies and trade winds, are stronger when located further south in most of the centuries and vice versa. Furthermore, the northward shift and weakening of the westerlies during the cooler Little Ice Age (LIA) in the 16th century in these simulations is consistent with evidence from sediment cores for the same period (Granger et al., 2018), of a weak Agulhas leakage during periods of cooler climate.

Nevertheless, model simulations as well as proxies hold uncertainties. Model simulations tend to overestimate the response to external climate forcing in the Southern Hemisphere during the last two millennia (PAGES 2k-PMIP3 group, 2015). Proxy analyses found that the period of the LIA has a different spatial and temporal structure on both hemispheres (PAGES 2k Network, 2013). The scarcity of temporally highly resolved climate archives (e.g. tree rings) induce large non-climatic noise into southern hemispheric reconstructions, also leading to increased uncertainty, too (PAGES 2k-PMIP3 group, 2015). Despite these uncertainties, the temperature levels have in general been lower during the period of the LIA in the Southern Hemisphere for both, proxy (Granger et al., 2018) and model simulations.

– *Projected trends for the 21th century:* The future evolution of the wind stress was analyzed with three emission scenarios of the MPI-ESM, differing in the prescribed strength of greenhouse gas forcing. Future trends of rcp2.6, the weakest scenario, indicate a northward shift and weakening of the westerlies, while trends derived from scenario rcp8.5, the strongest scenario, indicate a southward shift and strengthening. The scenario with moderate emissions, rcp4.5, has no significant trend. As ozone concentration changes are equal in the three scenarios, these different trajectories lead us to

conclude that the ozone pool will be the dominant driver, if greenhouse gas emissions are drastically reduced. In the more probable business-as-usual scenario (rcp8.5) the increase in atmospheric greenhouse gas concentrations will override the ozone effect and will lead to a further poleward shift and strengthening of the westerlies.

Such a poleward shift and intensification of the westerlies will enhance the Agulhas leakage and, therefore, cause a larger

transport of warm Indian Ocean water into the South Atlantic. This is likely to affect the Benguela upwelling system in several ways, and may have done so in the recent and geological past. A stronger leakage may increase the volume and changes the properties of the Eastern South Atlantic Central Water (ESACW), containing Agulhas water that enters the Benguela upwelling system (Tim et al., 2018). This water mass will become younger, warmer, richer in oxygen, and its higher volume increases the share of preformed nutrients in the South Benguela system. In contrast, a weakened

Agulhas leakage due to a northward shift of the wind systems reduces the contribution of Agulhas water in the upwelling region, possibly to be compensated by the inflow of an older intermediate water mass (the South Atlantic Central Water (SACW) dominant in the northern Benguela upwelling system) with low oxygen, high $CO_2$ and nutrient concentrations. We postulate that variability in the positions and strengths of the dominant wind systems - remote westerlies and the trade winds - set the pace for the Agulhas contribution to the upwelling feed water masses in the Benguela upwelling

system, impacting the productivity of the region, its $CO_2$ balance, ecosystems, and living natural resources.

*Data availability.* NCEP1 is available at $https : //www.esrl.noaa.gov/psd/data/gridded/data.ncep.reanalysis.derived.surfaceflux.html$, JRA-55 at $https : //rda.ucar.edu$, ERA20C at $https : //www.ecmwf.int/en/forecasts/datasets/reanalysis-datasets/era-20c$, and COREv2 at $https : //data1.gfdl.noaa.gov/nomads/forms/core/COREv2/CIAF\_v2.html$. The MPI-ESM-MR, MPI-ESM-LR, and ECHAM6XR ($doi : 10.1594/WDCC/CLISAP\_MPI-ESM-XR\_t255l95$) are available at the CERA data base

($https : //cera-www.dkrz.de/WDCC/ui/cerasearch/$). The INAL20 data set and ECHAM2k data set are available upon request by Franziska Schwarzkopf and Sebastian Wagner, respectively.

*Author contributions.* NT and EZ conducted the analysis. FUS performed the INALT20 simulation. All authors participated in the discussion of the results and the writing of the paper.

*Competing interests.* The authors declare that they have no conflict of interest.

*Acknowledgements.* This work received funding from the Federal Ministry of Education and Research of Germany within the SPACES-Agulhas II project (grants 03F0750A and 03F0750C) and the SPACES-CASISAC project (grants 03F0796A and 03F0796D). The model simulation INALT20 has been performed at the North-German Supercomputing Alliance (HLRN). Our thanks goes to Sebastian Wagner for his advices on paleoclimate reconstructions and for providing the MPI-ESM-P simulation of the past two millennia (ECHAM2k).

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
