# Peer review of "Analysis of position and strength of westerlies and trades and implications for Agulhas leakage and South Benguela upwelling"

_Earth System Dynamics, 2019_

## Referee Comment (RC1) · Anonymous Referee #1 · 23 May 2019

General comment

I appreciate reading this study as it is important to further study the impact of the SH wind system on Agulhas leakage dynamics and Benguella upwelling system –past, present, future. In this respect, I think it is important to highlight in the community the results here that reinforce some earlier findings in that the westerlies strength is important in driving leakage dynamics in comparison to older, outdated theories that are centered around the "width of the gateway" on the southern tip of Africa.

Abstract

Page 1 Rephrase to: Line 3: Agulhas leakage constitutes a fraction of warm and saline

water transport from the Indian Ocean into the South Atlantic.

Line 4: "The leakage is stronger during intensified westerlies and probably also when the wind systems are shifted poleward." Probably? If you are not sure or there is no evidence based on that study for that I would leave it out. Line 10: Give numbers here for the $CO_2$ emission scenarios or the RCP ones you referring to. Rephrase: Line 15: An increased contribution of Agulhas water to the upwelling system will feed water masses that will import more preformed nutrients and oxygen into the upwelling region. Line 19: with larger scale implications –like what? Line 20: change to Southern Hemisphere Westerlies and Easterly Trade winds Line 20: Here, we analyze several observational . . .. . .the last century and past two millennia. With the aim to understand what? Page 2 Line 28: As the Peeters et al. 2004 record is based on qualitative reconstructions of Agulhas leakage rather than quantitative numbers I would suggest to rephrase that to: "During glacial periods leakage was strongly diminished, based on qualitative reconstructions of foraminiferal assemblage counts, whereas the transport of Indian Ocean waters into the South Atlantic was enhanced during interglacial periods (Peeters et al., 2004).

Line 1: Simon et al. 2013 and 2015 actually, noted that changes in temperature and salinity in the Agulhas leakage is at least partly the result of variability in the composition in the current itself and can be a poor indicator of the strength of the leakage. Hence please rephrase that part to actually refer to the citations in an appropriate way.

Line 2: I dislike the "gateway theory" of driving AL amount very much. The common assumption is that shifts of the Southern Hemisphere westerly wind belt, (in particular the position of the zero wind stress curl) would have led to the widening/narrowing of the gap between Africa and the STF, thereby controlling the amount of warm salty Indian Ocean waters leaking into the South Atlantic. However, this assumption has been questioned (De Boer et al., 2013; Durgadoo et al., 2013). These studies showed

that the position of the STF is not related to the position/shifts in the wind belt i.e., position of the zero wind stress curl and that Agulhas leakage increases with northward shifted westerlies a scenario originally proposed for a narrower gateway. It is therefore unclear whether shifts of the wind fields did in fact act to alter past rates of Agulhas leakage, which might imply that other factors, despite the movement of the STF, were equally important in determining leakage.

Page 7: Line 1-6: Here the work of Loveday, B. R., P. Penven, and C. J. C. Reason (2015), Southern Annular Mode and westerly-wind-driven changes in Indian-Atlantic exchange mechanisms, Geophys. Res. Lett., 42, 4912–4921, doi:10.1002/2015GL064256. should be cited and discussed in comparison. Page 9: Line 6: Peeters et al. 2004 can't be used as reference for the LIA comparison. Moreover, ther are more studies in the area that cover the LIA interval and should be taken into account here when comparing to data. e.g. Hahn et al., 2017 Clim. Past, 13, 649–665, 2017 https://doi.org/10.5194/cp-13-649-2017 Moreover, if westerlies shifted equatorward and or weakened during glacials remains debated and speculative until now.

Line 13: In the weaker emission scenario, by contrast, significant trends mark a northward shift of the westerlies and a weakening of trades and westerlies.

So I wonder how the different RCP scenarios can provide such different results and hence how reliable they are then at all? If the models are struggling to reproduce the trends in the observational time period how can we believe any estimate for the RCP scenarios? Moreover, I don't understand the explanation given for the differences? Here more explanation would be appreciated with the regards to the ozone recovery mentioned.

Page 11 Fig.6: That is interesting result. Hence looking at the Agulhas Current itself it seems like that more a northerly position of the trades is linked to positive SST anomalies in the current itself but actually the opposite for the areas outside the main flow

path. How is a northerly position of the trades related to warmer Agulhas SSTs in the model? That part is a bit confusing to start with in terms of which ocean areas around South Africa are correlating with what position of the trades? Page 13 Line 9: Here another perspective should be give as Beal& Elipot 2016 showed based on observations that there is a broadening not strengthening of the Agulhas Current since the early 1990s.

---

## Referee Comment (RC2) · Anonymous Referee #2 · 23 Jun 2019

This article claims to be about the Benguela upwelling region (and Agulhas leakage), but I find it to be mostly on analyses of wind stress strengths and position variability in data products and simulations, over historical climates as well as future scenarios. There are many interesting aspects and results in the paper, but I find them to be poorly presented and hard to follow. Often, the results are not explicitly shown in figures. For example an upwelling index is discussed but never shown. "Upwelling" even appears in the title but I think it does not receive much attention. There isn't even a figure that focuses on the Benguela region. The correlation with SST seems interesting, but it is only shown for one simluation. The different simulations, historical and future on top of data products are difficult to follow and receive only individual attention. In the

figures often only one simluation or dataset is shown and it is diffiult to get a grip on the comprehensive analysis and consistency, or not, over different historical and future scenarios (exception is table 1). This paper should be rewritten in a more structured way and figures should represent the results of all experiments and datasets, otherwise the number of simulations or time periods should be limited. If not a more quantitative analysis on upwelling and Agulhas leakage (with aspects different than already considered in other papers) is done, the subject is more something like "An analysis of westerly and trade winds strengths and position over the South Atlantic and Indian ocean in historical and future climates"..

Abstract: use present tense in abstract and distinguish what was done in other studies before and what the present article investigates.

Please include a figure at the beginning to introduce the broader region, the circulations, winds, processes involved and indicate the areas where you take what kind of averages.

page 2, line 28: what do you mean by "parallel" (two lines can be parallel) ? Change wording.

page 4, line 10: dependency —> variation

page 4, line 27-30: including the ocean velocities in the wind stress calculations could include a feedback in the coupled climate model simulation, is that important?

page 5, line 3-5: Confusing, there are four datasets, but you show only one in Fig 1 ?

page 6, line 7: contrarily, change word

It is difficult to keep track of the different averaging regions, at first sight it seems confusing, the difference between wind stress trend Fig 1 and 3. Fig 1 there is a decreasing trend, Fig 3 increasing.

The wind stress analysis could be presented in a more compact way (combine Fig 1-3

as subpanels in one Figure and describe in caption the specifics.

Fig 4 and temperature gradient analysis: I don't think this adds any interesting information. Given the dominance of geostrophy, there is bound to be a correlation between temperature gradients and wind stress. Why is this analysis important at this point?

section 3.2: change to variability in past and future climate

page 9, lines 1-7: what do you mean by "calculated for each century separately"? Why not show the curves? Unclear.

page 10, line 13: "our" analysis

page 12, line 18: due to geostrophy, strong SAT gradients are naturally correlated with strong winds, that is not a driving mechanism.

Your conclusions are very comprehensive, but a lot of them are never shown explicitly in the results. (The position of the winds is only weakly correlated with South Benguela upwelling intensity, In contrast, the strength of the trades is significantly correlated with Benguela upwelling, with more intense trades being linked to stronger upwelling in South Benguela.. and the whole paragraph).

page 13,line 9-12: these are conclusions from other studies, not from your results

page 14,line 13: "This is likely to affect the Benguela upwelling system in several ways,.." why don't you show it here, I thought this is the subject of the paper ?

---

## Author Comment (AC1) · 17 Jul 2019

Reviewer #1:

We thank the reviewer for the constructive report and recommendations to revise the manuscript. In the following, we list our responses to the comments and the changes that we would introduce in the manuscript.

Page 1 Rephrase to: Line 3: Agulhas leakage constitutes a fraction of warm and saline water transport from the Indian Ocean into the South Atlantic.

It would be changed as suggested

[Figure]

Line 4: "The leakage is stronger during intensified westerlies and probably also when the wind systems are shifted poleward." Probably? If you are not sure or there is no evidence based on that study for that I would leave it out.

It would be left out

Line 10: Give numbers here for the CO2 emission scenarios or the RCP ones you referring to.

RCPs would be added

Rephrase:Line 15: An increased contribution of Agulhas water to the upwelling system will feed water masses that will import more preformed nutrients and oxygen into the upwelling region.

It would be rephrased as suggegsted

Line 19: with larger scale implications –like what? Line 20: change to Southern Hemisphere Westerlies and Easterly Trade winds. Line 20: Here, we analyze several observational......the last century and past two millennia. With the aim to understand what?

This is rather a question of writing style. We use the first paragraph of the introduction to prelude the analyses we did. As we analyze the winds here, the aim to understand the impact on upwelling and leakage is stated in the last paragraph of the introduction. Paragraph changes to: "The regional oceanographic phenomena around Southern Africa, the Agulhas Current, the Agulhas leakage, and the Benguela upwelling, are all three to a large extent influenced by one of the two wind regimes in this region, the Southern Hemisphere Westerlies and the Easterly Trade winds. Thus, here, we analyze several observational data sets and model simulations."

Page 2 Line 28: As the Peeters et al. 2004 record is based on qualitative reconstructions of Agulhas leakage rather than quantitative numbers I would suggest to rephrase that to: "During glacial periods leakage was strongly diminished, based on qualitative reconstructions of foraminiferal assemblage counts, whereas the transport of Indian

Ocean waters into the South Atlantic was enhanced during interglacial periods (Peeters et al., 2004).

It would be changed as suggested

Page 3 Line 1: Simon et al. 2013 and 2015 actually, noted that changes in temperature and salinity in the Agulhas leakage is at least partly the result of variability in the composition in the current itself and can be a poor indicator of the strength of the leakage. Hence please rephrase that part to actually refer to the citations in an appropriate way.

We would exclude this sentence when reformulating this paragraph. See the new wording in the answer to the following remark.

Line 2: I dislike the "gateway theory" of driving AL amount very much. The common assumption is that shifts of the Southern Hemisphere westerly wind belt, (in particular the position of the zero wind stress curl) would have led to the widening/narrowing of the gap between Africa and the STF, thereby controlling the amount of warm salty Indian Ocean waters leaking into the South Atlantic. However, this assumption has been questioned (De Boer et al., 2013; Durgadoo et al., 2013). These studies showed that the position of the STF is not related to the position/shifts in the wind belt i.e.,position of the zero wind stress curl and that Agulhas leakage increases with northward shifted westerlies a scenario originally proposed for a narrower gateway. It is therefore unclear whether shifts of the wind fields did in fact act to alter past rates of Agulhas leakage, which might imply that other factors, despite the movement of the STF, were equally important in determining leakage.

We would change the paragraph about the impact of the position of the westerlies according to the reviewers suggestion: "The impact of the position of the westerlies on the Agulhas leakage is still under debate. The studies of DeRuijter et al. (1982) and of Biastoch et al. (2009) found that a more northerly position hinders the westward flow of Agulhas water into the South Atlantic, whereas a more southerly position leads to a wider passage of throughflow between the south coast of South Africa and the

westerlies. However, recent studies found diverging results. De Boer et al. (2013) showed the there is no linkage between the position of the zero wind stress curl and of the subtropical front. The study by Durgadoo et al. (2013) found that an enhanced leakage tends to occur when westerlies are shifted equatorwards due to the redistribution of momentum input by the winds. Nevertheless, there is consensus on the impact of the westerlies on the Agulhas leakage and that the strength of westerlies is the key driver of the leakage: stronger westerlies lead to a stronger wind stress curl and an intensified transport from the Indian Ocean into the South Atlantic (Durgadoo et al., 2013 and Cheng et al., 2018)."

Page 7: Line 1-6: Here the work of Loveday, B. R., P. Penven, and C. J.C. Reason (2015), Southern Annular Mode and westerly-wind-driven changes inIndian-Atlantic exchange mechanisms, Geophys.Res.Lett., 42, 4912–4921,doi:10.1002/2015GL064256. should be cited and discussed in comparison.

We would add this reference and discuss it: "These results agree with the study of Loveday et al. (2015). They found that SAM modulates Indian Ocean westerlies and further detected the impact of this connection to the Agulhas leakage."

Page 9: Line 6: Peeters et al. 2004 cant be used as reference for the LIA comparison. More-over, ther are more studies in the area that cover the LIA interval and should be taken into account here when comparing to data. e.g. Hahn et al., 2017 Clim. Past, 13,649–665, 2017 https://doi.org/10.5194/cp-13-649-2017 Moreover, if westerlies shifted equatorward and or weakened during glacials remains debated and speculative until now.

We would modify this part as followed: "This is supported by the studies by Hahn et al. (2017), Stager et al. (2012), and Granger et al. (2018), who also found that the winds shifted equatorward during cooler and poleward during warmer periods. Nevertheless, the position of Southern Hemisphere westerlies during the LIA is still under debate and seems to depend on the region as studies of the African, the South American and the

Australian sector provide varying results (Chase and Meadows, 2007)."

Line 13: In the weaker emission scenario, by contrast, significant trends mark a northward shift of the westerlies and a weakening of trades and westerlies. So I wonder how the different RCP scenarios can provide such different results and hence how reliable they are then at all? If the models are struggling to reproduce the trends in the observational time period how can we believe any estimate for the RCP scenarios? Moreover, I dont understand the explanation given for the differences?Here more explanation would be appreciated with the regards to the ozone recovery mentioned.

We would modify and extend our explanation: "Although the simulations struggle to reproduce the observed trends, we provide here an explanation as to why the simulations with different scenarios of greenhouse emissions produce different trends of the wind systems. This explanation involves the compensation of diverging trends caused by ozone and by greenhouse gases. The strongest scenario rcp8.5 indicate a poleward shift and intensification of westerlies (and a poleward shift and weakening of the trades). In the weaker emission scenario (rcp2.6), by contrast, the simulation displays a northward shift of the westerlies and a weakening of trades and westerlies. As prescribed ozone concentrations are the same in all three scenarios, the amount of greenhouse gas emissions is likely the factor that causes the difference in the simulated trends of the wind systems. It has been previously found that the the ozone recovery causes a northward shift and a weakening of the tropospheric jet, and a lowering of the SAM (Southern Annular Mode) values Watson et al. (2012). This is likely the effect seen in the simulations with the weaker rcp2.6 scenario. Only with the stronger rcp8.5 scenario are the emissions strong enough to counteract the effects of ozone recovery The simulated trends under rcp4.5 forcing are not insignificant , which in our interpretation would indicate a balance of both driving factors, ozone recovery and greenhouse gas emissions. It has to be kept in mind that these results depend on the model ."

Page 11: Fig.6: That is interesting result. Hence looking at the Agulhas Current itself it seems like that more a northerly position of the trades is linked to positive SST anomalies in the current itself but actually the opposite for the areas outside the main flow path. How is a northerly position of the trades related to warmer Agulhas SSTs in the model? That part is a bit confusing to start with in terms of which ocean areas around South Africa are correlating with what position of the trades?

We suppose that a northerly position of the trades reduces upwelling in the western part of the south coast and that the Agulhas Current is located closer to the coast. Hence, the positive correlation at the coast is linked to the more northerly position of the Agulhas Current and the warmer SSTs due to reduced upwelling. The negative correlation is where the current is located when trades are located more southerly. To indicate the selected regions in this study, we would add a figure early in the manuscript showing them.

Page 13 Line 9: Here an-other perspective should be give as Beal& Elipot 2016 showed based on observations that there is a broadening not strengthening of the Agulhas Current since the early1990s.

We would add and change this to: "Beal et al. (2016) confirms that intensified winds impact the Agulhas Current, though not by strengthening it but by broadening it. Thus, the coherent variability and trends in the two wind systems cause a modulation of both oceanic components, Agulhas Current and Agulhas leakage."

---

## Author Comment (AC2) · 17 Jul 2019

Reviewer #2:

We thank the reviewer for the time and effort invested in reviewing the manuscript and for the useful comments and suggestions. In the following, we list our responses to those comments and describe how we would change the manuscript according to these recommendations.

This article claims to be about the Benguela upwelling region (and Agulhas leakage), but I find it to be mostly on analyses of wind stress strengths and position variability

in data products and simulations, over historical climates as well as future scenarios. There are many interesting aspects and results in the paper, but I find them to be poorly presented and hard to follow. Often, the results are not explicitly shown in figures. For example an upwelling index is discussed but never shown. "Upwelling" even appears in the title but I think it does not receive much attention. There isnt even a figure that focuses on the Benguela region. The correlation with SST seems interesting, but it is only shown for one simluation. The different simulations, historical and future on top of data products are difficult to follow and receive only individual attention. In the figures often only one simluation or dataset is shown and it is diffiult to get a grip on the comprehensive analysis and consistency, or not, over different historical and future scenarios (exception is table 1). This paper should be rewritten in a more structured way and figures should represent the results of all experiments and datasets, otherwise the number of simulations or time periods should be limited. If not a more quantitative analysis on upwelling and Agulhas leakage (with aspects different than already considered in other papers) is done, the subject is more something like "An analysis of westerly and trade winds strengths and position over the South Atlantic and Indian ocean in historical and future climates"..

To reduce the focus on the Benguela upwelling system we would change the title to "Analysis of position and strength of westerlies and trades and their impact on the Agulhas leakage and the South Benguela Upwelling". The new title emphasizes that we analyze the westerlies and trades with the purpose of detecting changes in the Agulhas leakage and the Benguela upwelling system. To indicate the selected regions in this study, we would add a figure early in the manuscript showing them. Furthermore, we would add the correlation coefficients regarding the Benguela upwelling system in the text and of any other time series correlation if not already mentioned. However, we are a somewhat skeptical about the utility of showing all figures for all data sets. Adding that many subplots would not be beneficial for the manuscript clarity.

Abstract: use present tense in abstract and distinguish what was done in other studies

before and what the present article investigates.

We would change to present tense and distinguish more clearly between our results and results of previous studies.

Please include a figure at the beginning to introduce the broader region, the circulations, winds, processes involved and indicate the areas where you take what kind of averages.

We would include such a figure in the revised manuscript.

page 2, line 28: what do you mean by "parallel" (two lines can be parallel) ? Change wording.

With "parallel" we mean "in conjunction with". We would change to this wording in the revised manuscript.

page 4, line 10: dependency → variation

We would change it accordingly.

page 4, line 27-30: including the ocean velocities in the wind stress calculations could include a feedback in the coupled climate model simulation, is that important?

In the case of INALT20 it is not relevant because the simulation is not coupled. The wind stress in the coupled millennium simulations is calculated by the coupled model itself, we do not calculate it a posterior.

page 5, line 3-5: Confusing, there are four datasets, but you show only one in Fig 1 ?

We only show one data set in the figure as all data sets show a similar trend and it would not be an improvement to show the others too. We will make the reader more clearly aware of this.

page 6, line 7: contrarily, change word

We would change wording to "The correlations are numerically negative because winds

are positive when directed eastward."

It is difficult to keep track of the different averaging regions, at first sight it seems confusing, the difference between wind stress trend Fig 1 and 3. Fig 1 there is a decreasing trend, Fig 3 increasing. The wind stress analysis could be presented in a more compact way (combine Fig 1-3 as subpanels in one Figure and describe in caption the specifics.

Figure 1 shows the latitudinal position and figure 3 shows the strength of the wind stress. We would combine both figures in the revised manuscript.

Fig 4 and temperature gradient analysis: I dont think this adds any interesting information. Given the dominance of geostrophy, there is bound to be a correlation between temperature gradients and wind stress. Why is this analysis important at this point?

We added this section to show that the increase in temperature gradient in the past has contributed to the changes in the wind stress. Furthermore, this correlation hints at a possible tendency for future development of the wind systems.

section 3.2: change to variability in past and future climate

We would change it as suggested.

page 9, lines 1-7: what do you mean by "calculated for each century separately"? Why not show the curves? Unclear.

As there has been warmer and colder centuries, we calculate the trend for each century separately to investigate if the wind stress shifted poleward and intensified simultaneously as well as shifted equatorward and weakened. We would add a figure of the trends.

page 10, line 13: "our" analysis

Sure. This is a typo.

[Figure]

page 12, line 18: due to geostrophy, strong SAT gradients are naturally correlated with strong winds, that is not a driving mechanism.

This question is subtle and depends on which data set is being analyzed. In the case of meteorological reanalysis, where the SST are prescribed, the causality can only go from the prescribed SSTs, and therefore SAT over ocean, to the simulated winds. In these cases, one can indeed refer to a driver if a strong statistical link is found. In the case of coupled simulations, the question is not as clear. However, there is a very plausible mechanisms by which temperature gradients drive the geostrophic part of the westerlies through the effect of temperature on density. A mechanism by which zonal winds may physically cause meridional temperature gradients is not as plausible. Nevertheless, we will explain this two situations in the revised version.

Your conclusions are very comprehensive, but a lot of them are never shown explicitly in the results. (The position of the winds is only weakly correlated with South Benguela upwelling intensity, In contrast, the strength of the trades is significantly correlated with Benguela upwelling, with more intense trades being linked to stronger upwelling in South Benguela.. and the whole paragraph).

We would add correlation coefficients in the section 3.3 to support our conclusions and show our results more explicitly. The rest of conclusions in this paragraph can be seen in figure 6.

page 13,line 9-12: these are conclusions from other studies, not from your results

This section is the discussions and conclusions section. Therefore, the compare our results with other studies and put them into context. When referring to other studies, like in this sentences, these studies are cited. We will make the separation between our results and previous finding more clear in the revised manuscript.

page 14,line 13: "This is likely to affect the Benguela upwelling system in several ways,.." why dont you show it here, I thought this is the subject of the paper ?

This is not the content of this paper and has been already published in another paper (as cited in the manuscript). The new title clarifies the subject of the manuscript.

---

## Author Response (AR1)

Authors response:

We thank both reviews for the constructive comments and suggestions. We revised our manuscript with regard to their comments and present our responses and changes made in the manuscript in the following:

**Reviewer #1:**
Abstract
- Rephrase to: Line 3: Agulhas leakage constitutes a fraction of warm and saline water transport from the Indian Ocean into the South Atlantic.
  **Changed as suggested.**
- Line 4: "The leakage is stronger during intensified westerlies and probably also when the wind systems are shifted poleward." Probably? If you are not sure or there is no evidence based on that study for that I would leave it out.
  **We left out the second part of the sentence.**
- Line 10: Give numbers here for the CO2 emission scenarios or the RCP ones you referring to.
  **RCPs have been added now.**
- Rephrase:Line 15: An increased contribution of Agulhas water to the upwelling system will feed water masses that will import more preformed nutrients and oxygen into the upwelling region.
  **Rephrased to:**
  ***„An increased contribution of Agulhas water to the upwelling water masses will import more preformed nutrients and oxygen into the upwelling region."***
- Line 19: with larger scale implications –like what? Line 20: change to Southern Hemisphere Westerlies and Easterly Trade winds. Line 20: Here, we analyze several observational......the last century and past two millennia. With the aim to understand what?
  **This is rather a question of writing style. We use the first paragraph of the introduction to prelude the analyses we did. As we analyze the winds here, the aim to understand the impact on upwelling and leakage is stated in the last paragraph of the introduction. Paragraph changed to:** "***The regional oceanographic phenomena around Southern Africa, the Agulhas Current, the Agulhas leakage, and the Benguela upwelling, are all three to a large extent influenced by one of the two wind regimes in this region, the Southern Hemisphere westerlies and the easterly trade winds. In this study, we analyze several observational data sets and model simulations to understand the variability and trends in the intensity and spatial distribution of westerly and trade winds in the South Atlantic-Indian Ocean over the past few decades, the last century and past two millennia. We also compare observed trends with trends projected in future climate simulations driven, among others, by changing greenhouse gas and ozone concentrations.***"

- Line 28: As the Peeters et al. 2004 record is based on qualitative reconstructions of Agulhas leakage rather than quantitative numbers I would suggest to rephrase that to: "During glacial periods leakage was strongly diminished, based on qualitative reconstructions of foraminiferal assemblage counts, whereas the transport of Indian Ocean waters into the South Atlantic was enhanced during interglacial periods (Peeters et al., 2004)***.***
  **Changed as suggested.**

- Line 1: Simon et al. 2013 and 2015 actually, noted that changes in temperature and salinity in the Agulhas leakage is at least partly the result of variability in the composition in the current itself and can be a poor indicator of the strength of the leakage. Hence please rephrase that part to actually refer to the citations in an appropriate way.
  **We left out this sentence by reformulating this paragraph. See the new wording in the answer to the following remark.**
- Line 2: I dislike the "gateway theory" of driving AL amount very much. The common assumption is that shifts of the Southern Hemisphere westerly wind belt, (in particular the position of the zero wind stress curl) would have led to the widening/narrowing of the gap between Africa and the STF, thereby controlling the amount of warm salty Indian Ocean waters leaking into the South Atlantic. However, this assumption has been questioned (De Boer et al., 2013; Durgadoo et al., 2013). These studies showed that the position of the STF is not related to the position/shifts in the wind belt i.e.,position of the zero wind stress curl and that Agulhas leakage increases with northward shifted westerlies a scenario originally proposed for a narrower gateway. It is therefore unclear whether shifts of the wind fields did in fact act to alter past rates of Agulhas leakage, which might imply that other factors, despite the movement of the STF, were equally important in determining leakage.
  **We changed the paragraph about the impact of the position of the westerlies according to the reviewer's suggestion (now page 2, line 33): "*The impact of the position of the westerlies on the Agulhas leakage is still under debate. The studies of DeRuijter et al. (1982) and Biastoch et al. (2009) found that a more northerly position hinders the westward flow of Agulhas water into the South Atlantic, whereas a more southerly position leads to a wider passage of throughflow between the south coast of South Africa and the westerlies. However, recent studies found diverging results. De Boer et al. (2013) showed that there is no linkage between the position of the zero wind stress curl and of the subtropical front. The study by Durgadoo et al. (2013) found that an enhanced leakage tends to occur when westerlies are shifted equatorwards due to the redistribution of momentum input by the winds.*
  *Nevertheless, there is consensus on the impact of the westerlies on the Agulhas leakage, with the strength of westerlies being the key driver of the leakage: stronger westerlies lead to a stronger wind stress curl and an intensified transport from the Indian Ocean into the South Atlantic (Durgadoo et al., 2013 and Cheng et al., 2018)."***

Page 7:
- Line 1-6: Here the work of Loveday, B. R., P. Penven, and C. J.C. Reason (2015), Southern Annular Mode and westerly-wind-driven changes in Indian-Atlantic exchange mechanisms, Geophys.Res.Lett., 42, 4912–4921,doi:10.1002/2015GL064256. should be cited and discussed in comparison.
  **We added this reference and discussed it (now line 14): *"These results agree with the study of Loveday et al. (2015). They found that SAM modulates Indian Ocean westerlies and further detected the impact of this connection to the Agulhas leakage."***

Page 9:

- Line 6: Peeters et al. 2004 can't be used as reference for the LIA comparison. Moreover, there are more studies in the area that cover the LIA interval and should be taken into account here when comparing to data. e.g. Hahn et al., 2017 Clim. Past, 13,649–665, 2017 https://doi.org/10.5194/cp-13-649-2017 Moreover, if westerlies shifted equatorward and or weakened during glacials remains debated and speculative until now.
  **We modified this part as follows (now page 10, line 3): *"This is supported by the studies by Hahn et al. (2017), Stager et al. (2012), and Granger et al. (2018), who also found that the winds shifted equatorward during cooler and poleward during warmer periods. Nevertheless, the position of Southern Hemisphere westerlies during the LIA is still under debate and seems to depend on the region, as studies of the African, the South American and the Australian sector provide varying results (Chase and Meadows, 2007)."***
- Line 13: In the weaker emission scenario, by contrast, significant trends mark a northward shift of the westerlies and a weakening of trades and westerlies. So I wonder how the different RCP scenarios can provide such different results and hence how reliable they are then at all? If the models are struggling to reproduce the trends in the observational time period how can we believe any estimate for the RCP scenarios? Moreover, I don't understand the explanation given for the differences? Here more explanation would be appreciated with the regards to the ozone recovery mentioned.
  ***We modified and extended our explanation (now page 10, line 12): "Although the simulations of the historical period produce weaker trends than observed, we here provide an explanation as to why the simulations with different scenarios of greenhouse emissions produce different trends in the wind systems. This explanation involves the compensation of diverging trends caused by ozone on the one hand and by anthropogenic greenhouse gases on the other hand. The strongest scenario rcp8.5 indicates a poleward shift and intensification of westerlies (and a poleward shift and weakening of the trades). In the weaker emission scenario (rcp2.6), by contrast, the simulation displays a northward shift of the westerlies and a weakening of trades and westerlies. As prescribed ozone concentrations are the same in all three scenarios, the amount of anthropogenic greenhouse gas emissions is likely the factor that causes the difference in the simulated trends in the wind systems. It has previously been found in simulations that the ozone recovery causes a northward shift and a weakening of the tropospheric jet and a lowering of the SAM (Southern Annular Mode) values (Watson et al., 2012). This is likely the effect seen in the simulations with the weaker rcp2.6 scenario. Only with the stronger rcp8.5 scenario the emissions are strong enough to counteract the effects of ozone recovery. The simulated trends under rcp4.5 forcing are not statistically significant, which in our interpretation would indicate a balance of both driving factors, ozone recovery and anthropogenic greenhouse gas emissions. It has to be kept in mind that details of these results depend on the model."***

Page 11:
- Fig.6: That is interesting result. Hence looking at the Agulhas Current itself it seems like that more a northerly position of the trades is linked to positive SST anomalies in the current itself but actually the opposite for the areas outside the main flow path. How is a northerly position of the trades related to warmer Agulhas SSTs in the model? That part is a bit confusing to start with in terms of which ocean areas around South Africa are correlating with what position of the trades?

We suppose that a northerly position of the trades reduces upwelling in the western part of the south coast and that the Agulhas Current is located closer to the coast. Hence, the positive correlation at the coast is linked to the more northerly position of the Agulhas Current and the warmer SSTs due to reduced upwelling. The negative correlation is where the current is located when trades are located more southerly.

**To indicate the selected regions in this study, we added a figure early in the manuscript showing them (new Figure 1).**

- Line 9: Here another perspective should be give as Beal& Elipot 2016 showed based on observations that there is a broadening not strengthening of the Agulhas Current since the early1990s.

    **We added the reference and changed this to (now page 14, line 5):** *"Beal et al. (2016) confirm that intensified winds impact the Agulhas Current, though not by strengthening it but by broadening it. Thus, the coherent variability and trends in the two wind systems cause a modulation of both oceanic components, Agulhas Current and Agulhas leakage."*

Reviewer #2:

This article claims to be about the Benguela upwelling region (and Agulhas leakage), but I find it to be mostly on analyses of wind stress strengths and position variability in data products and simulations, over historical climates as well as future scenarios. There are many interesting aspects and results in the paper, but I find them to be poorly presented and hard to follow. Often, the results are not explicitly shown in figures. For example an upwelling index is discussed but never shown. "Upwelling" even appears in the title but I think it does not receive much attention. There isn't even a figure that focuses on the Benguela region. The correlation with SST seems interesting, but it is only shown for one simulation. The different simulations, historical and future on top of data products are difficult to follow and receive only individual attention. In the figures often only one simulation or dataset is shown and it is difficult to get a grip on the comprehensive analysis and consistency, or not, over different historical and future scenarios (exception is table 1). This paper should be rewritten in a more structured way and figures should represent the results of all experiments and datasets, otherwise the number of simulations or time periods should be limited. If not a more quantitative analysis on upwelling and Agulhas leakage (with aspects different than already considered in other papers) is done, the subject is more something like "An analysis of westerly and trade winds strengths and position over the South Atlantic and Indian ocean in historical and future climates"..

**To reduce the focus on the Benguela upwelling system we changed the title to "Analysis of position and strength of westerlies and trades and their impact on the Agulhas leakage and the South Benguela upwelling". The new title emphasizes that we analyze the westerlies and trades with the purpose of detecting changes in the Agulhas leakage and the Benguela upwelling system.**

**To indicate the selected regions in this study, we added a figure early in the manuscript showing them (new Figure 1).**

**Furthermore, we added the correlation coefficients regarding the Benguela upwelling system in the text and of any other time series correlation if not already mentioned.**

**However, we are somewhat skeptical about the utility of showing all figures for all data sets in the manuscript itself. We added the combined figure of 1**

**and 3 (now figure 2) of the other data sets and the time series of position and strength of the westerlies and trades for each century from the ECHAM6_2k simulation in a supplement.**

Abstract: use present tense in abstract and distinguish what was done in other studies before and what the present article investigates.

**We changed it to present tense and distinguish more clearly between our results and results of previous studies.**

Please include a figure at the beginning to introduce the broader region, the circulations, winds, processes involved and indicate the areas where you take what kind of averages.

**We include such a figure in the new version of the manuscript (new Figure 1).**

page 2, line 28: what do you mean by "parallel" (two lines can be parallel) ? Change wording.

**With "parallel" we mean "in conjunction with". We changed to "in conjunction with" in the revised manuscript.**

page 4, line 10: dependency —> variation

**We changed it accordingly (now page 5, line 4).**

page 4, line 27-30: including the ocean velocities in the wind stress calculations could include a feedback in the coupled climate model simulation, is that important?

**In the case of INALT20 it is not relevant because the simulation is not coupled. The wind stress in the coupled millennium simulations is calculated by the coupled model itself, we do not calculate it a posteriori.**

page 5, line 3-5: Confusing, there are four datasets, but you show only one in Fig 1 ?

**We only show one data set in the figure in the manuscript itself as all data sets show a similar trend. We make the reader more clearly aware of this and added figures of the other data sets in the supplement.**

page 6, line 7: contrarily, change word

**We changed wording to (now line 32) „The correlations are numerically negative because winds are positive when directed eastward."**

It is difficult to keep track of the different averaging regions, at first sight it seems confusing, the difference between wind stress trend Fig 1 and 3. Fig 1 there is a decreasing trend, Fig 3 increasing. The wind stress analysis could be presented in a more compact way (combine Fig 1-3 as subpanels in one Figure and describe in caption the specifics.

**Figure 1 shows the latitudinal position and figure 3 shows the strength of the wind stress. We combined both figures in the revised manuscript (new Figure 2).**

Fig 4 and temperature gradient analysis: I don't think this adds any interesting information. Given the dominance of geostrophy, there is bound to be a correlation between temperature gradients and wind stress. Why is this analysis important at this point?

**We added this section to show that the increase in temperature gradient in the past has contributed to the changes in the wind stress. Furthermore, this**

**correlation hints at a possible tendency for future development of the wind systems when the temperature gradient may change in the future due to the greenhouse gas radiative forcing. The correlation between the temperature gradient and the wind stress is not 1, which implies that other factors also impact the link between the temperature gradient and the wind stress. We think that to include the strength of this link is a useful piece of information.**

section 3.2: change to variability in past and future climate
> **We changed it as suggested.**

page 9, lines 1-7: what do you mean by "calculated for each century separately"? Why not show the curves? Unclear.
> **As there has been warmer and colder centuries, we calculate the trend for each century separately to investigate if the wind stress shifted poleward and intensified simultaneously as well as shifted equatorward and weakened. We added a figure of the trends in the supplementary.**

page 10, line 13: "our" analysis
> **Sure. This is a typo (now page 11, line 2).**

page 12, line 18: due to geostrophy, strong SAT gradients are naturally correlated with strong winds, that is not a driving mechanism.
> **This question is subtle and depends on which data set is being analyzed. We explain this in the revised version (now page 13, line 11):**
> *"As for these atmospheric data sets that are generated in simulations where the SSTs are prescribed, the causal link between the temperature gradient and the wind stress can only come from the prescribed SSTs to the simulated wind stress. Thus, these correlations indeed result from a physical link. In the case of coupled simulations, the answer is not as clear. However, there is a very plausible mechanism by which temperature gradients drive the geostrophic part of the westerlies through the effect of temperature on density. A mechanism by which zonal winds may physically cause meridional temperature gradients is not as plausible".*

Your conclusions are very comprehensive, but a lot of them are never shown explicitly in the results. (The position of the winds is only weakly correlated with South Benguela upwelling intensity, In contrast, the strength of the trades is significantly correlated with Benguela upwelling, with more intense trades being linked to stronger upwelling in South Benguela.. and the whole paragraph).
> **We added correlation coefficients in section 3.3 to support our conclusions and show our results more explicitly. The rest of the conclusions in this paragraph can be seen in figure 6.**

page 13,line 9-12: these are conclusions from other studies, not from your results
> **This section is the discussions and conclusions section. Therefore, we compare our results with other studies and put them into context. When referring to other studies, like in this sentences, these studies are cited. We made the separation between our results and previous finding clearer in the revised manuscript.**

page 14,line 13: "This is likely to affect the Benguela upwelling system in several ways,.." why don't you show it here, I thought this is the subject of the paper ?

**This is not the content of this paper and has been already published in another paper (as cited in the manuscript). The new title clarifies the subject of the manuscript.**

[revised manuscript text omitted]

---

## Author Response (AR2)

Thank you again for evaluating the revised version. We have accepted the suggestion of the referee and have changed the title accordingly.